# Aptamer-Based Fluorescent Biosensor for the Rapid and Sensitive Detection of Allergens in Food Matrices

**DOI:** 10.3390/foods10112598

**Published:** 2021-10-27

**Authors:** Liping Hong, Mingfei Pan, Xiaoqian Xie, Kaixin Liu, Jingying Yang, Shan Wang, Shuo Wang

**Affiliations:** 1State Key Laboratory of Food Nutrition and Safety, Tianjin University of Science and Technology, Tianjin 300457, China; honglpstu@163.com (L.H.); qianxx8135@163.com (X.X.); Liukx2019@163.com (K.L.); yangjy0823@126.com (J.Y.); wshan0929niu@163.com (S.W.); s.wang@tust.edu.cn (S.W.); 2Key Laboratory of Food Nutrition and Safety, Ministry of Education of China, Tianjin University of Science and Technology, Tianjin 300457, China

**Keywords:** allergen, detection, aptamer, fluorescence, food safety

## Abstract

Food allergies have seriously affected the life quality of some people and even endangered their lives. At present, there is still no effective cure for food allergies. Avoiding the intake of allergenic food is still the most effective way to prevent allergic diseases. Therefore, it is necessary to develop rapid, accurate, sensitive, and reliable analysis methods to detect food allergens from different sources. Aptamers are oligonucleotide sequences that can bind to a variety of targets with high specificity and selectivity, and they are often combined with different transduction technologies, thereby constructing various types of aptamer sensors. In recent years, with the development of technology and the application of new materials, the sensitivity, portability, and cost of fluorescence sensing technology have been greatly improved. Therefore, aptamer-based fluorescence sensing technology has been widely developed and applied in the specific recognition of food allergens. In this paper, the classification of major allergens and their characteristics in animal and plant foods were comprehensively reviewed, and the preparation principles and practical applications of aptamer-based fluorescence biosensors are summarized. In addition, we hope that this article can provide some strategies for the rapid and sensitive detection of allergens in food matrices.

## 1. Introduction

Food allergies, an adverse reaction to antigenic substances in food mediated by the immune system, have been recognized as a global health issue with increasing prevalence in the field of food safety [1,2]. Most food allergies are immunoglobulin (Ig) E-mediated type I (immediate type) hypersensitivity reactions [3]. An epidemiological survey by the institute of infectious diseases shows that about 6–9.3% of children and 3.4–5.0% of adults have food allergies, which means the incidence of food allergies in infants and children is generally higher than that of adults [4,5,6]. However, there is still no standard cure for food allergies except avoiding eating foods that contain allergens. Therefore, the development of rapid and effective detection methods for allergens in food matrices is a topic of concern in the whole society.

In the past few decades, many mature techniques have been widely used in the detection of food allergens, such as the enzyme-linked immunosorbent assay (ELISA), liquid chromatography-mass spectrometry (LC-MS), and polymerase chain reaction (PCR) [7,8,9]. The ELISA method has been widely used in the detection of food allergens due to its high specificity and sensitivity. Nevertheless, due to the influence of various external conditions such as food processing methods, there would be false positive and false negative results [10,11]. Moreover, PCR method is usually used for monitoring allergic components in food processing due to its high specificity and high automation. However, PCR technology is not suitable for identifying allergen proteins with unascertained genes, which limits its scope of application [12,13]. Furthermore, HPLC and LC-MS are standard strategies for the quantitative analysis of allergens in various food matrices. Because of the precision requirements of the instruments, these methods usually require strict sample pre-treatment processes, a larger sample volume, and a longer analysis time, resulting in a higher detection cost [14]. Currently, biosensors with high sensitivity and specificity, such as surface-enhanced Raman spectroscopy (SERS), electrochemical biosensors, and quartz crystal microbalance (QCM) biosensors, can rapidly analyze and screen food allergens and allow on-site analysis, which are considered effective detection technology [15,16,17]. However, these biosensors usually require expensive instruments, proficient operators, and higher requirements for the surrounding environment. Therefore, there is an urgent need to develop rapid, accurate, sensitive, and easy-to-operate detection methods to quantify allergens in food matrices.

Nucleic acid aptamer is a nucleic acid sequence that can specifically recognize the target, screened by systematic evolution of ligands by exponential enrichment (SELEX) in vitro [18]. The combination of aptamer and target is achieved through single-stranded oligonucleotide deoxyribonucleic acid (DNA) or ribonucleic acid (RNA) folded into a specific three-dimensional structure (stem-loop, hairpin and G-quadruplex and other spatial conformations) [19,20,21]. Regardless of the technical requirements for the preparation of aptamers, the convenience and timeliness far exceed those of antibodies. Moreover, the screened aptamers can be artificially synthesized, which is easy to achieve standardization. In recent years, aptamers have received extensive attention due to their veracity, high specificity, and affinity, and they have been used in disease diagnosis and treatment, drug delivery, food safety testing, and environmental monitoring [22,23,24]. In terms of food safety, the application of aptamers to the detection of allergens in food matrices is expected to achieve the goal of accurate, rapid, and low-cost detection of allergens.

Fluorescence detection technology, due to its low cost, high sensitivity, simple performance, has attracted wide attention [25,26,27]. Combining fluorescence detection technology with aptamers, the development of biosensors with high sensitivity and simple detection procedures provides a feasible strategy for the detection of food allergens [28]. Aptamer-based fluorescence sensing detection is a relatively common analysis method. The fluorophore is combined with the aptamer in a labeled or non-labeled manner, and the analyte concentration and other information are reflected by the interaction of the excitation light and the identification element [29,30]. Furthermore, fluorescence intensity, decay rate, spectral properties, and fluorescence anisotropy can be used alone or in combination as signal detection means. Therefore, the method has the advantages of rapid, accurate, multi-targets, high sensitivity, simple instrumentation, and in-situ detection [31,32]. This review introduces the allergens in different food matrices in detail, explains the preparation principle of aptamer fluorescence sensing technology and its application in the detection of allergens in food matrices, and looks forward to the future development of food allergen detection methods.

## 2. Classification of Food Allergens

A great variety of food allergens exist widely in nature. According to the source of food allergens, they can be classified into animal allergens, plant allergens, and fungal allergens. Since the occurrence of fungal allergens is not very common, only animal and plant allergens are discussed in this review. Table 1 lists the classification of major food allergens, allergy symptoms, and other information. Specific information about food allergies is also discussed in the following sections.

### 2.1. Common Allergens in Animal Food

#### 2.1.1. Seafood

Seafood refers to several different groups of edible aquatic animals, including fish, crustaceans, and molluscs [57,58]. For culinary reasons, the two invertebrate groups of crustaceans and mollusks are usually combined into shellfish [59,60]. Seafood are one of the “eight major allergens” identified by the Food and Agriculture Organization of the United Nations (FAO) in 1995 [61]. Seafood allergy is a serious global public health problem with increasing prevalence, which affects 2.5% of the global population due to the rapid increase in consumption [62,63]. The exposure pathways of seafood allergy include ingestion, contact, and even inhalation [64]. However, the main method of sensitization is through ingestion after cooking or processing [65].

Among fish allergenic proteins, parvalbumin (PV, Pan h 1) belongs to one of the Ca^2+^ binding proteins and can maintain allergenicity under severe destructive conditions (such as heat, chemical denaturation, and proteolytic enzymes). α-Lineage and β-lineage are two homologous lineages of PV, in which β-lineage is the major allergen and is present in almost all teleost fishes [66,67]. Furthermore, different types of PV in fish have highly conserved structures and cross-reactivity. Indeed, a study has shown that 95% of fish allergy sufferers are allergic to PV [68]. Except for PV, aldolase (40 kDa), enolase (47–50 kDa), vitellogenin, collagen, and tropomyosin (TM, Cra c 1) are minor allergens of fish [69]. In recent years, with the deepening of studies on fish allergic diseases, many new fish allergens have been reported. In fish muscle, six proteins with molecular weights between 25 and 57 kDa, such as triose phosphate isomerase (28 kDa) and pyruvate kinase (57 kDa), have been confirmed to have Ig E binding activity, but the sensitization of these new allergens needs further study [70,71].

At present, there are many allergens in shellfish products have been identified. TM, the first major allergen found in shellfish food, which protein spatial structure is a highly stable α-helical dimer. Ig E specific binding experiments showed that 72–98% of patients are allergic to TM [72,73]. Arginine kinase (AK, Cra c 2) is also identified as the major allergen in crustacean aquatic product muscle. There were 10–51% of shrimp allergic patients exhibit positive IgE binding to purified AK. Studies have shown that although the spatial structure of AK is more complex than that of TM, damage to the spatial structure (treated by heat and pH) can reduce its sensitization [74,75,76]. In addition to the two major allergens mentioned above, the allergen sarcoplasmic calcium binding protein in the muscles of crustacean aquatic products has an EF hand structure similar to that of PV, which can induce 29–50% of shrimp allergic patients to be positive for Ig E binding [77]. In recent years, some new allergens, such as myosin light chain, troponin, triose phosphate isomerase, and paramyosin, which can cause mild allergic reactions, have also been isolated and identified [78,79]. However, the sensitizing properties of these proteins remain unclear and further studies are needed.

#### 2.1.2. Milk

Milk allergy seriously affects the life of people allergic to milk and has a relatively high incidence worldwide [80]. Relevant studies have shown that about 2–7% of infants and children suffer from milk allergies due to the ingestion of milk and milk products [81]. We need to clarify the structure and sensitization mechanism of milk allergens, thereby establishing accurate and sensitive detection and analysis methods to help people prevent the occurrence of allergic diseases. The main allergens in milk are casein (Bos d 8), β-lactoglobulin (Bos d 5) and α-lactalbumin (Bos d 4) [82,83].

Casein accounts for about 80% of milk protein and exists in milk in the form of micelle, which belongs to a large class of calcium-binding phosphoproteins [84]. It is coded by different genes located on the same chromosome and is divided into four types: αs1-casein, αs2-casein, β-casein, and κ-casein, which account for 32%, 10%, 28%, and 10%, respectively [85]. The difference in structure and content of casein in human milk and cow’s milk is the main cause of casein sensitization. A study has shown that about 65% of sufferers are allergic to casein [86]. Whey protein accounts for 20% of milk protein, and the main allergenic components are α-lactalbumin and β-lactoglobulin [87]. Compared with casein, whey protein has a higher secondary and tertiary structure due to the fact that they are not phosphorylated and contain intramolecular disulfide bonds, which will make them resistant to acids or enzymes, so they can pass the intestinal mucosa smoothly, be recognized by the immune cells in the human body, and trigger an immune response. Studies have shown that about 27.6–62.8% of allergic people are caused by α-lactalbumin, and about 82% are caused by β-lactoglobulin [88,89,90].

#### 2.1.3. Egg

Eggs are the second leading cause of food allergies, accounting for 35% of infants and young children and 12% of adults [91,92,93]. However, the symptoms of egg allergy may gradually disappear with age [94]. At present, there are four major allergens found in egg whites, including ovomucoid (OVM, Gal d1, 11%), ovalbumin (OVA, Gal d2, 54%), ovotransferrin (OVT, Gal d3, 12%), and lysozyme (Lys, Gal d4, 3.5%), which account for nearly 80% of the total egg white protein. α-Livetin (Gal d5) and yolkglycoprotein42 (Gal d6) are the major allergens in the yolk [95,96]. Most egg allergies are caused by allergens in egg white [97].

OVM is composed of 186 amino acid residues and contains three functional domains with independent homologous structures. It is reported that the third functional domain of OVM has the strongest allergenicity [98]. The 20–25% glycosyl component contained in the structure makes OVM very stable to the thermal processing and enzymatic hydrolysis of trypsin [99]. OVA is a monomeric water-soluble protein composed of 385 amino acid residues and is the most abundant protein in egg whites [100]. At present, OVA has been widely used as a model protein to study protein structure, functional properties and food allergy animal models [101]. There are reports in the literature that OVA is a stronger allergen than other egg allergens [102]. OVT, a glycoprotein, is a monoglycated polypeptide composed of 686 amino acid residues. OVT possess many biological activities due to the N-terminal and C-terminal two domains each contain a Fe^3+^ binding site, respectively [103]. In addition, OVT is very similar to serum transferrin, which easy to form stable complexes with metal ions [104]. Lys, a weak allergen, is an alkaline globulin that consists of 129 amino acid residues in a single polypeptide [105]. At present, studies on Lys mainly focus on its antibacterial properties. Although OVM and OVA are more common allergens than Lys, the high usage rate of Lys in food and drug production makes it one of the important egg white allergens in allergy research [106].

### 2.2. Common Allergens in Plant Food

#### 2.2.1. Peanut

Peanuts are one of the more common food allergens, which often cause severe allergic reactions. Peanut allergens have high thermal stability, acid and enzymatic resistance, and the general production method cannot remove the allergenicity [107,108]. In the actual production process, food processing often requires complex production processes, which will cause cross-contamination between foods. Therefore, it is difficult to accurately determine whether some foods contain peanut allergens [109,110]. Among the 13 allergens identified in peanuts, Ara h 1, Ara h 2, and Ara h 3 are the major allergens in peanut [111]. Ara h 1, a soluble protein, accounts for approximately 12–16% of the total peanut protein, which belongs to the pea globule protein and can cause more than 90% of allergic reactions. In the natural state, Ara h 1 has two kinds of monomers and trimers, which exist in the form of soluble proteins. For the trimeric form of Ara h 1, it is a homotrimeric glycoprotein formed by connecting three monomers by hydrophobic interaction. Most epitopes are more or less hidden in the natural Lys trimer complex, which can protect the monomer from degradation, and will lead to increased allergies. Guillon et al. determined the stability of the Ara h 1 trimer structure and described its spatial structure in detail [112]. Ara h 2 belongs to blue bean protein and is also the main peanut allergen, with the content of 5.9–9.3% of the total peanut protein, which can also cause more than 90% of allergic reactions. As there are many disulfide bonds in Ara h 2, the structure is very stable [113]. In addition, Ara h 2 contains two genetic variants, Ara h 2.01 (16.7 kDa) and Ara h 2.02 (18 kDa). Ara h 3 is 62–72% similar to glycinin and also has two allogeneic proteins, Ara h 3.01 (60 kDa) and Ara h 3.02 (37 kDa), in which the serum of more than 44% of peanut allergy patients can recognize Ara h 3.01 [114,115].

#### 2.2.2. Wheat

As one of the three major grains and the staple foods of mankind, wheat is widely cultivated all over the world [116,117]. However, wheat is also a major plant allergen, and some people have severe allergic reactions to wheat [118,119,120,121]. Although most wheat allergies can cause mild reactions, in some cases, it can be life-threatening. The protein in wheat can be divided into water-soluble albumin, salt-soluble globulin, and gluten soluble in ethanol or acid according to solubility, and the allergens in wheat mostly come from glutelin [122,123]. At present, most of the wheat allergens that have been identified belong to the α-amylase/trypsin inhibitor family, and there are 18 species: Tri a 12, Tri a 14, Tri a 17–21, Tri a 25–28, Tri a 36–37, and Tri a 41–45. Among them, Tri a 14, Tri a 19, and Tri a 36 are the major allergens [124,125,126].

#### 2.2.3. Soybean

Soybean, a major plant protein source, which is extensively used in the food processing industry. Simultaneously, soybean is also one of the “eight major allergens” that identified by the FAO [127]. Studies have shown that with the increase in the use of soybean and soybean products, about 1–6% of infants and children are allergic to soybean and the incidence of soybean allergies in adults is also increasing [128,129]. There are 11 kinds of allergenic proteins found in soybeans, namely: β-conglycinin (Gly m 5), hydrophobic protein, defensive protein, inhibitory protein, SAM22, 7S globulin, glycinin (Gly m 6), 2S albumin, lectin, lipoxidase, and trypsin inhibitor [130,131,132]. Among them, Gly m Bd 28K and Gly m Bd 30K of the 7S globulin and the α-subunit Gly m Bd 60K of β-conglycinin are the main allergens in soybeans.

Gly m Bd 30K (34 kDa) is a water-insoluble monomolecular glycoprotein consisting of 257 amino acid residues, which can be combined with subunits of β-conglycinin through disulfide bonds to participate in the folding of soy protein [133]. Gly m Bd 28K (26 kDa) is a broad bean globulin-like protein belonging to the Cupin superfamily and contains 220 amino acid residues [134]. β-conglycinin contains three subunits, namely α- subunit (68 Ku), α’-subunit (71 Ku), and β-subunit (50 Ku), and they exist in the form of homologous or heterologous trimers. Among them, Gly m Bd 60K is very stable, which is formed by the combination of polysaccharide and aspartic acid at the N-terminus of protein [135].

#### 2.2.4. Nuts

Nuts mainly include almonds, cashews, walnuts, hazelnuts, pistachios, Brazil nuts, etc., which can cause allergic reactions [136]. Generally used as seeds or fruits, most nut proteins belong to three conservative seed storage proteins, including 2S albumin, 7S legumin and 11S legumin. 2S albumin belongs to the group of prolamins, which have the characteristics of low molecular weight and multiple cysteine residues in their sequence. Furthermore, most allergens in the prolamin group are highly resistant to heat, pH, and gastrointestinal enzymes due to their small and compact structure [137,138]. The 7S legumin is usually a trimeric protein, which has weak stability due to lack of disulfide bonds. The 11S legumin is a mature trimeric protein, and both subunits are connected by disulfide bonds. Moreover, both 7S legumin and 11S legumin belong to the Cupin family [139].

## 3. Application of Aptamer-Based Fluorescence Biosensors in the Detection of Different Food Allergens

### 3.1. Nucleic Acid Aptamer Screening Procedure

Aptamers are obtained through in vitro screening from random DNA or RNA libraries. The screening procedure of aptamers mainly includes the steps of library establishment, incubation, isolation, amplification, single-strand preparation, and purification [140,141]. After several rounds of repeated screening, aptamers with high affinity and high specificity can be obtained. The specific procedure is shown in Figure 1 [142]. (1) To construct a chemically synthesized oligonucleotide library, each oligonucleotide molecule usually contains about 80 nucleotides, including random sequences and constant primers. (2) The constructed library is incubated with the target, and part of the target is bound to the oligonucleotide. (3) Elution and separation of unbound or weakly bound oligonucleotides determines the screening efficiency of SELEX. (4) PCR or real-time PCR is used to amplify the combined sequence. (5) The amplified sequence obtained from the reaction is used to prepare the next round of secondary library.

### 3.2. Aptamer-Target Interaction Mechanisms

The interaction between an aptamer and its target is the core of the SELEX process and practical application. The nature of this interaction depends on the type and size of the target [143]. When the target is a small molecule, the aptamer can integrate the target into its structure through stacking (with flat, aromatic ligands and ions), electrostatic complementation (with oligosaccharides and charged amino acids), and/or hydrogen bonding interactions [144]. This interaction with small molecules gives the aptamer higher specificity that can distinguish two molecules that differ by only one methyl group, which may be due to steric hindrance. When the target is a protein macromolecule, contrary to the above situation, the aptamer will be integrated into the structure of its target or attached to the surface. Since proteins often exhibit a high degree of structural complexity, the interaction mechanism between aptamers and proteins is more diverse than that of small molecules [145,146]. In addition to hydrogen bond interactions, polar interactions and structural complementarity are also included [147]. Among them, RNA- or DNA-binding motifs that exhibit this structural complementarity are often found in nature, including helical motifs, leucine zippers, homologous domains, and beta-sheet motifs.

In addition, the nature of the interaction depends not only on the type and size of the target, but also on the structural complexity of the aptamer itself. According to the difference of sequence, aptamers may assume a polymeric state through the formation of G-tetrachromes or i-motifs, both of which can lead to interactions between multiple oligonucleotides [148]. Interestingly, according to the principle of “induced fit”, the formation of aptam-target complex may involve conformational changes of the target, the aptamer, or both [149]. This principle leads to better shape complementarity, which in turn promotes stronger hydrogen bonding and van der Waals forces. Furthermore, the charge on the surface of the target also affect the interaction between the aptamer and its target. The negative charges will weaken or even prevent the binding with the aptamer, because they will adversely interact with the negatively charged phosphate groups contained in DNA and RNA. On the other hand, a positive charge can enhance the strength of the interaction, but it may also aggravate the occurrence of non-specific binding [150,151,152].

### 3.3. Detection of Animal Food Allergens

Seafood allergy is not only an important public health issue, but a serious food safety issue that affects the quality of life and may even be life threatening [64]. For people with seafood allergies, avoiding foods containing seafood allergens is still the best option. Therefore, the monitoring of allergens is a process that requires strict supervision [153]. In order to evaluate seafood allergens, new detection methods with high sensitivity and high efficiency are required.

As we all know, magnetic separation is easy to operate and can effectively reduce or eliminate the interference from complex matrices in food. Therefore, based on functionalized magnetic nanoparticles (MNPs) as a separation carrier, Zhang et al. developed a simple and versatile label-free aptamer-based fluorescent sensor for the sensitive detection of TM (Figure 2a) [154]. In the study, OliGreen dye was selected as a fluorescent signal probe. The aptamer hybridizes with the capture probe bound to the surface of the MNPs to form an aptamer-MNPs complex as detection probe. When interacting with the target, the conformation of the complex changes, resulting in the release of the aptamer from the surface of the MNPs. So, the released aptamer in the supernatant produced a significant fluorescence enhancement signal, which is because the combination of OliGreen dye and ssDNA will produce ultrasensitive and specific fluorescence enhancement phenomenon. It is worth noting that when the commercially available OliGreen dye is in the free state, the fluorescence is weak or no fluorescence, but the fluorescence will increase by more than 1000 times once combined with the aptamer ssDNA. Under the optimal conditions, the linear range was 0.4–5 μg mL^−1^ (R^2^ = 0.996), with a limit of detection LOD of 77 ng mL^−1^. In addition, the highly selective aptamer-based fluorescent sensor was successfully applied to the detection of TM in food matrix. Wu et al. also developed a similar sensor with a LOD of 4.2 nM and the concentration linear from 0.5–50 μg mL^−1^ [155]. Recently, Chinappan et al. developed an aptamer-based fluorescent-labeled sensor for the detection of TM. (Figure 2b) [156]. Graphene oxide (GO) is used as a platform for screening the minimum length of aptamer sequences that can bind to the target with high affinity. A fluorescein dye labeled GO quenches the truncated aptamer by π-stacking and hydrophobic interactions. After the addition of TM, the fluorescence was restored due to the competitive binding of the aptamer to GO. More importantly, the aptamer selected in this study is a truncated ligand fragment, which has four times higher affinity than the full-sequence aptamer, with a LOD of 2.5 nM. The developed aptamer-based fluorescence sensor can complete the detection within 30 min. The performance of the sensor was confirmed in the addition experiment of chicken broth, and a high percentage recovery rate (~97 ± 10%) was achieved. Compared with the above studies, the sensitivity and specificity of this work have been greatly improved.

Fluorescence resonance energy transfer (FRET) is a mechanism widely used in the preparation of biosensors, which is an energy transfer phenomenon between two fluorescent molecules that are very close [157]. Zhou et al. designed an aptamer-based “on-off-on” fluorescent biosensor based on FRET and used developed carboxyl functionalized carbon quantum dots (cCQDs) and GO for the detection of shellfish allergen arginine kinase (AK) (Figure 3A) [158]. The cCQDs-aptamer probe and GO self-assemble for the first time through a specific π-π interaction, so that the fluorescence of cCQDs is effectively quenched. After the addition of AK, cCQDs-aptamer is released from the GO surface and then forms the cCQDs-aptamers-AK complex, which restores the fluorescence of cCQDs. The aptamer-based FRET sensor can perform sensitive detection in the AK concentration range of 0.001–10 μg mL^−1^, with a LOD of 0.14 ng mL^−1^ (S/N = 3) and a limit of quantification (LOQ) of 0.27 ng mL^−1^ (S/N = 10). Furthermore, in a control experiment with a blank sample, it was found that the sensor has high specificity. This reliable, precise, highly specific, and easy-to-operate aptamer sensor may provide a new perspective for the application of fluorescence sensing technology in the field of food safety.

In recent years, biosensors based on dual signals or functions have received widespread attention due to the diversity of detection. Dual-mode nanosensors usually use colorimetric and fluorescent reporters to achieve convenient visual inspection and highly sensitive fluorescent detection [160,161]. Wang et al. developed a dual-mode aptamer-based fluorescent sensor for the detection of PV, the major allergen of fish (Figure 3B) [159]. In Figure 3C(a), aptamer towards PV was obtained by in vitro screening of random ssDNA library containing a 40-mer randomized region using the triple-mode GO-SELEX. The aptamer-modified gold nanoparticle (AuNP-APT), complementary short-strand modified gold nanoparticles (AuNP-CS1), and fluorescent dye-labeled complementary short-strands (FAM-CS2) were assembled by DNA hybridization. After the addition of PV, the competitive interaction with aptamer leads to the decomposition of the aptamer sensor, resulting in the color shift of the AuNPs solution and the recovery of the FAM-CS2 fluorescence signal. The results showed that the aptamer sensor showed a good colorimetric response (2.5–20 μg mL^−1^) and linear fluorescence correlation (2.38–40 μg mL^−1^) in the PV concentration range. In addition, the affinity and specificity of the aptamer sensor were also investigated, as shown in Figure 3C(b,c). Therefore, aptamer 5 with good affinity (KD = 7.66 × 10^−7^ M) and specificity is the best aptamer for aptasensor construction. They also studied the feasibility of aptamer sensor in real fish samples, revealing the potential in field of monitoring and quantitative detection of food allergens.

Recently, as an alternative to antibodies, the use of peptide aptamers as biosensors has attracted more attention. Peptide aptamers usually contain 10–20 amino acids, which the high selective recognition ability is equivalent to that of antibodies. Phadke et al. used ribosome display technology to select two fluorescent peptide aptamers Cas1 and Cas2. for the detection of α-casein [162]. Among them, 7-nitrobenzofurazan (NBD)-modified aminophenylalanine is coupled to the translated peptides to prepare fluorescent peptide aptamers. This is because the peptide can quench the fluorescence of NBD. Once the peptide recognizes the target α-casein, the NBD-modified phenylalanine is released, and its fluorescence will instantly increase. It is worth noting that although the fluorescence of the two aptamers increased slightly in the presence of the control protein β-lactoglobulin, the modification of Cas1 with polyethylene glycol (PEG-Cas1) inhibited this phenomenon, which is because PEG-Cas1 may inhibit the interaction between aptamer and β-lactoglobulin. The aptamer sensor with a LOD of 0.04 mM, is equivalent to that of the kit. Moreover, the system can detect α-casein in short time (20–25 s) when compared with the 15 min required by immunochromatography kits. In addition, it is found that when using PEG-Cas1 to detect casein, the instant increase in fluorescence can be observed even with the naked eye. This study has contributed to improving the specificity of aptamers.

In order to reduce the incidence of milk allergy, hypoallergenic formula (HF) has been commercialized as a substitute for milk [163]. Nevertheless, in some cases, infants who consume these formula milk powder still have allergic reactions because of residual β-lactoglobulin in HF [164]. Therefore, it is necessary to establish a method that can detect the lower concentration of β-lactoglobulin. Shi et al. used carbon dots (CDs) as a fluorescent signal and Fe_3_O_4_ NPs as a magnetic separator to establish a fluorescent-labeled assay for the detection of β-lactoglobulin [165]. The assay is based on the hybridization between aptamers immobilized on Fe_3_O_4_ NPs and CDs-labeled complementary oligonucleotides (cDNA). In the presence of β-lactoglobulin, the aptamer preferentially binds to β-lactoglobulin, and part of the CDs-cDNA is released into the solution. After magnetic separation, the fluorescence signal of the supernatant increased with the increase of β-lactoglobulin concentration. Based on this, the aptamer assay with the range of 0.25–50 ng mL^−1^ and a LOD of 37 pg mL^−1^ has been successfully applied to the detection of trace β-lactoglobulin in HF. In the study of Qi et al., the binding mechanism of aptamer and β-lactoglobulin and the detection principle of fluorescent surface-enhanced Raman scattering (fluorescent-SERS) dual-mode aptamer sensor were thoroughly studied, which provides a theory basis and application potential for the development of aptamer sensors [166]. In Figure 4a, the circular dichroism of Lg-18, thermodynamic parameters analysis, secondary structure of Lg-18, and the result of molecular docking between aptamer Lg-18 and β-lactoglobulin were performed to illustrate the successful selection of aptamer. The specific response principle of the dual-mode aptamer sensor is shown in Figure 4b. The fluorescent-SERS aptamer sensor shows a wider linear range (10–5000 ng mL^−1^), and the LOD is 0.05 ng mL^−1^. Furthermore, under the interference of other proteins, the aptamer sensor showed excellent specificity.

Lys, as an allergen in egg and a biomarker of many diseases, its detection and quantification are of great significance in clinical diagnosis [167]. Sapkota et al. developed an aptamer sensor based on single-molecule FRET (smFRET) for the detection of Lys (Figure 5a) [168]. One of the arms has a blocking chain (B1), which is extended by 15 nucleotides to partially hybridize to the aptamer. The aptamer sensor remains open and almost no FRET efficiency occurs when Lys is not detected. After the addition of Lys, the aptamer binds to Lys and is displaced from the sensor, resulting in a foothold-mediated replacement of B1 by another chain, H1. At this time, the binding of Lys triggers the conformational transition state from low FRET to high FRET. Using this strategy, they demonstrated that the aptamer sensor can detect Lys at concentration as low as 30 nM, with a dynamic range extends to ~2 μM, and is almost free of interference from similar biomolecules. In addition, the smFRET method requires only a small number of aptamers, which offers the advantage of cost-effectiveness. In fluorescence detection, the presence of background fluorescence induced by ultraviolet-visible light in biological samples can lead to inaccurate detection results [169]. However, Ou et al. developed an X-ray nanocrystal scintillator aptamer sensor for sensitive detection of Lys based on the characteristics of weak scattering and almost no absorption of biological chromophores under X-ray irradiation (Figure 5b) [170]. In this study, aptamer-labeled lanthanide-doped nanocrystalline scintillators are designed to detect Lys quickly and sensitively through FRET. The use of low-dose X-rays as the excitation source and nanocrystals containing heavy atoms can achieve efficient luminescence, which endows the aptamer fluorescence sensor with high sensitivity (LOD: 0.94 nM), specificity, and sample recovery. In addition, this technology can provide a new generation of high-efficiency strategy without autofluorescence interference for the sensing and detection of biomarkers in biomedical applications.

### 3.4. Detection of Plant Food Allergens

Lupin, a legume plant, is widely cultivated around the world. α-Conglutin and β-conglutin are the main allergens of lupin, accounting for 33% and 45%, respectively [172,173]. FRET signals based on ligand-induced conformational changes of the aptamer, Mairal et al. developed a dimer-based dual fluorescent carrier-labeled biosensor using 11 mer truncated aptamers for the sensitive detection of β-conglutin [174]. According to reports, the 11-mer truncated aptamer can form a dimer structure in its natural state [175]. Based on this, the FRET probe was prepared by ligating two fluorophores with 11 mer truncated aptamers. The fluorophore Alexa Fluor 488 is excited at 492 nm and emits at 519 nm, and Alexa Fluor 555 is excited at 533 nm and emits at 568 nm. After the addition of β-conglutin, specific interactions can cause changes in the structure of the double aptamer, resulting in increased fluorescence at 568 nm. This method not only has the advantages of strong specificity, rapid detection, and high sensitivity (LOD: 150 pM), but can also be used for the direct detection of β-conglutin in food matrices at room temperature.

FRET based fluorescence detection methods are receiving increasing attention due to their high sensitivity. Weng et al. also used this mechanism to develop a quantum dot (Qdots) aptamer functionalized GO nano-biosensor integrated microfluidic system for the detection of peanut allergen Ara h 1 (Figure 5c) [171]. The nano-biosensor uses the Qdots-aptamer-GO complex as a probe, which will undergo conformational changes when interacting with Ara h 1. This is because the correlation constant between Qdots-aptamer and Ara h1 is greater than that between Qdots-aptamer and GO, which causes Qdots-aptamer to be released from GO, thereby restoring fluorescence. This one-step “turn on” detection with a LOD of 56 ng mL^−1^ in the ready-to-use microfluidic chip only needs 10 min to achieve the sensitive detection of Ara h 1. In addition, the use of small optical detectors to measure fluorescent signals improves the portability of the entire system and provides a promising method for the rapid, accurate, and economical on-site detection of other food allergens.

## 4. Conclusions and Prospective

In the field of food safety, food allergy is a worldwide problem. Therefore, the development of effective allergen quantitative detection technology is a problem that the food industry has been exploring. Benefiting from the development of aptamer technology, different types of aptamer sensors can be established by using the highly specific binding of aptamers and targets. In addition, the detection technology based on fluorescent signal has the advantages of no radioactivity, simple operation, high throughput, high sensitivity, and small sample size, which have attracted more and more attention. Therefore, fluorescent biosensors using aptamers as biorecognition ligands have made great progress in the sensitive, rapid, specific, and simple analysis of food allergens. Furthermore, fluorescent biosensors can be integrated with inexpensive and portable equipment, which provides a theoretical basis and practical application experience for the portable detection of food allergens. For food manufacturers, portable devices with appropriate sensitivity and specificity are the best choice for allergen testing during food processing. It is worth noting that the fluorescent biosensor allows to capture different allergens in food samples by changing different aptamer, which provides a strategy for the wide application of aptamer-based fluorescent sensors.

In recent years, aptamer-based fluorescent sensors have been successfully applied in the detection of allergens and other small molecules. Although aptamers have strong specificity to the target, when other analogs coexist or the concentration of the target is low, the influence of this weak non-specific signal cannot be ignored. Therefore, improving the level of specific aptamer screening technology is one of the future study directions of aptamer technology specificity. With the continuous efforts of scientific researchers, we believe that we will be able to gradually overcome the technical difficulties of aptamer fluorescence sensors in practical applications and make them more widely used in environmental detection, health, and medical fields.

## Figures and Tables

**Figure 1 foods-10-02598-f001:**
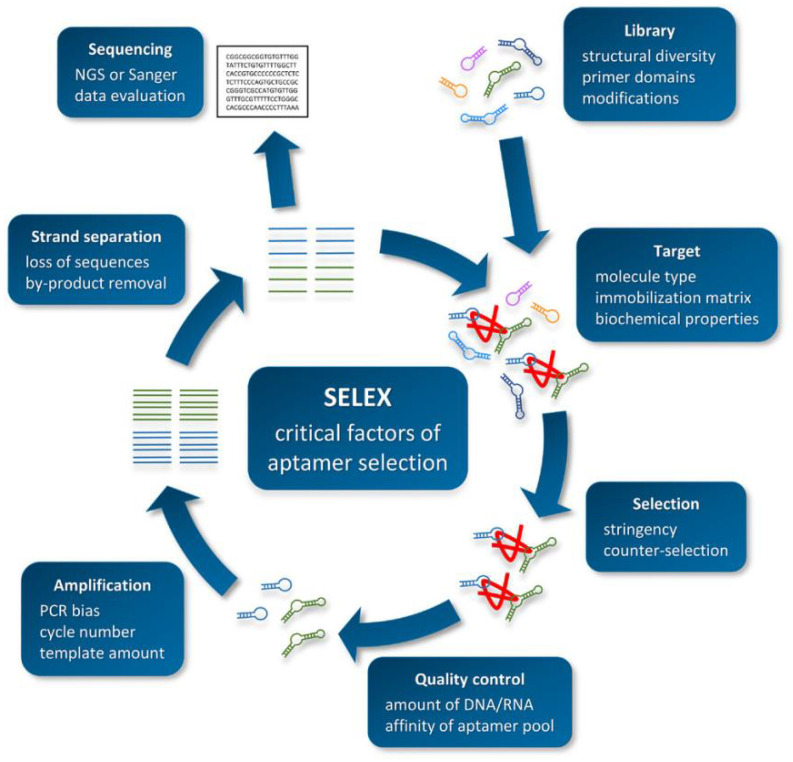
Schematic overview of the SELEX procedure. Reproduced with permission from [142]. Copyright Biotechnology and Applied Biochemistry, 2021.

**Figure 2 foods-10-02598-f002:**
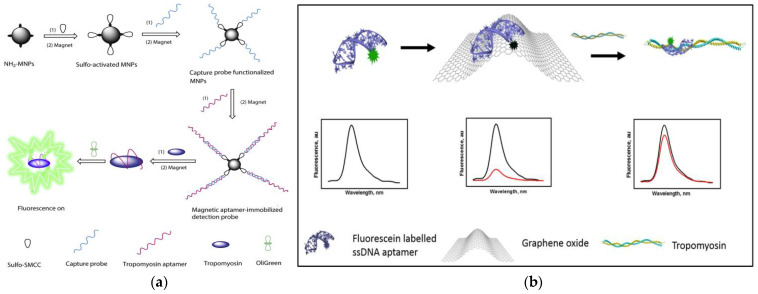
(**a**) Schematic of preparation of magnetic-assisted fluorescent aptamer for tropomyosin detection. Reproduced with permission from [154]. Copyright Sensors and Actuators B-Chemical, 2018. (**b**) Schematic of graphene oxide-based fluorescent aptamer biosensor for TM detection A: Changes in the fluorescence intensity of the aptamer released from the GO surface; B: The linear correlation of the fluorescence intensity of TMT2 (at 515 nm) with the concentration of TM. Reproduced with permission from [156]. Copyright Food Chemistry, 2020.

**Figure 3 foods-10-02598-f003:**
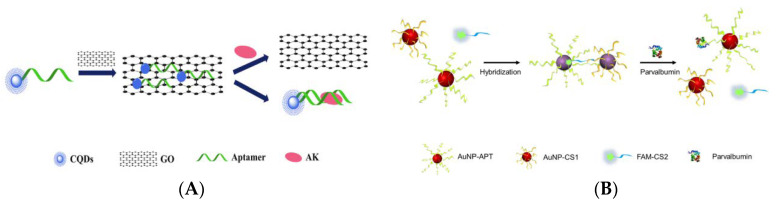
(**A**) Schematic of a “on-off-on” fluorescence aptasensor for AK detection. Reproduced with permission from [158]. Copyright Microchemical Journal, 2020. (**B**) Schematic of a dual-mode fluorescence sensor for PV detection based on AuNP color changes and FAM-CS2 fluorescence changes. (**C**) a: Schematic of the aptamer selection procedure by capturing GO-SELEX; b: Affinity of Apt5 towards PV; c: Specificity of Apt5 towards PV. Reproduced with permission from [159]. Copyright Microchemical Journal, 2020.

**Figure 4 foods-10-02598-f004:**
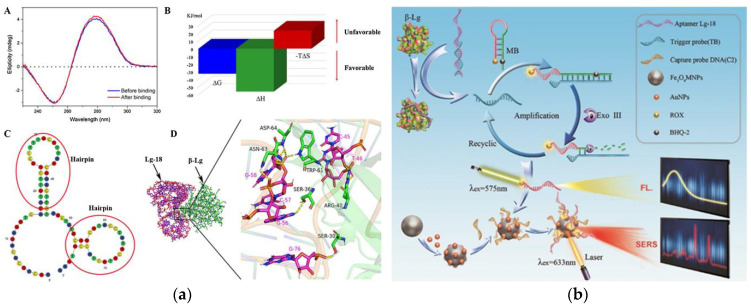
(**a**) Schematic of Lg-18 and β-lactoglobulin binding. A: Circular dichroism analysis of Lg-18 before and after binding. B: Analysis of thermodynamic parameters in the combined process. C: Secondary structure of Lg-18 predicted by Mfold online software. D: Analysis of molecular docking results of aptamer Lg-18 and β-lactoglobulin (**b**) Schematic of aptamer-based fluorescent Raman dual-mode biosensor for detection of β-lactoglobulin. Reproduced with permission from [166]. Copyright Sensors and Actuators B-Chemical, 2021.

**Figure 5 foods-10-02598-f005:**
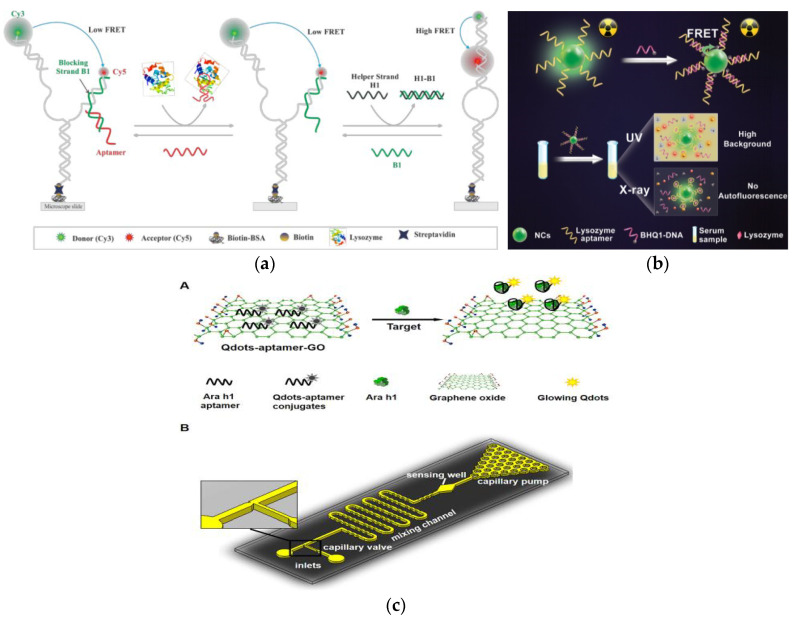
(**a**) Schematic of aptamer-based fluorescent biosensor for Lys detection. Reproduced with permission from [168]. Copyright sensors, 2020. (**b**) Schematic of aptamer sensor based on nanocrystal scintillator for detecting Lys without autofluorescence. Reproduced with permission from [170]. Copyright Analytical Chemistry, 2019. (**c**) A: Schematic of the Qdots-aptamer-GO quenching sensing principle; B: Schematic of the designed microfluidic chip. Reproduced with permission from [171]. Copyright Biosensors & Bioelectronics, 2016.

**Table 1 foods-10-02598-t001:** The major allergens in food matrices and their allergenic properties.

Food	Major Allergens	Molecular Mass (kDa)	Types of Proteins	The Structure of Proteins	Allergy Symptoms	Reference
Fish	Pan h 1	10–13	Calbindin	Contains 3 EF-hand regions (a motif composed of a 12-residue loop with a 12-residue-α-helix domain on each side), 2 of which can bind calcium.	Blushing, hives, nausea, stomach pain, and intestinal bleeding.	[33]
Shellfish	Cra c 1	33–39	Protein bound to actin	Adopting an α-helix structure, two molecules are entangled with each other to form a parallel dimeric α-helix structure.	Nausea, diarrhea, abdominal pain, and muscle paralysis.	[34]
	Cra c 2	38–45	Phosphoglycoprotein	Arginine kinase consists of an N-terminal domain (1–111) and a C-terminal domain (112–357). The N-terminal domain is all α-helices, and the C-terminal domain is an 8-strand anti-parallel β-sheet structure surrounded by 7 α-helices.		[35]
Milk	Bos d 8	57–37.5	Phosphate calcium binding protein	Consists of 4 independent proteins: αs1-casein, αs2-casein, β-casein, and κ-casein.	Skin rash, urticaria, eczema, vomiting, diarrhea, abdominal cramps, etc.	[36]
	Bos d 4	14.4	Combine with metal ions and participate in lactose synthesis	With a two-piece structure containing α-single loop and 310 helix larger subdomain.		[37]
	Bos d 5	18	Lipid transporter	Consists of two subunits connected by non-covalent bonds, mainly in the form of dimers.		[38]
Egg	Gal d1	28	Phosphoglycoprotein	Contains 3 independent homologous structural energy domains, and 3 functional domains are arranged consecutively in space.	Eczema, dermatitis, urticaria, vomiting, diarrhea, gastroesophageal reflux, etc.	[39]
	Gal d2	45	Phosphoglycoprotein	Containing 4 free sulfhydryl groups, composed of 385 amino acid residues, these amino acid residues are twisted and folded to form a spherical structure with high secondary structure, most of which are α-helix and β-sheet.		[40]
	Gal d3	77	Iron-binding glycoprotein	Consisting of 686 amino acids, including 12 disulfide bonds, the N-terminal and C-terminal 2 domains each contain a binding site for Fe^3+^.		[41]
	Gal d4	14.3	Basic globulin	A single peptide chain composed of 18 kinds of 129 amino acid residues, with 4 pairs of disulfide bonds to maintain the enzyme configuration, with lysine at the N-terminus and leucine at the C-terminus.		[42]
Peanut	Ara h 1	63.5	7S Globulin	The secondary structure contains β-turns, and the quaternary structure is a trimeric complex formed by 3 monomers.	Angioedema, hypotension, asthma, anaphylactic shock, etc.	[43]
	Ara h 2	17–20	2S Albumin	A monomeric protein.		[44]
	Ara h 3	57	11S Globulin	The N-terminal and C-terminal domains of the monomer form contain 2 ciupin folds (composed of two sets of parallel β-turns, random coils and 3 α-helices).		[45]
Wheat	Tri a 36	40	Gluten	-	Wheat exercise stimulates allergies, urticaria, dermatitis, bread asthma, nausea, and diarrhea.	[46]
Soybean	Gly m 5	150–200	7S Globulin	Trimer composed of α’-subunit, α-subunit and β subunit.	Red and itchy skin, asthma and allergic rhinitis, abdominal pain, diarrhea, etc.	[47]
	Gly m 6	320–360	11S Globulin	A hexamer composed of the interaction of G1, G2, G3, G4, and G5 subunits.		[48]
Nuts	Ana o 1	50	7S legumin	Exist as a trimer in natural state.	Metallic taste in the mouth, edema of the tongue or throat, difficulty breathing and swallowing, urticaria all over the body, flushing of the skin, cramping abdominal pain, nausea.	[49]
	Jug r 2	44		Consists of 593 amino acid residues.		[50]
	Cor a 11	48		Consists of 401 amino acid residues, with two potential N-glycosylation sites (Asn38 and Asn254) and a leader peptide of 46 amino acids.		[51]
	Ana o 3	14	2S albumin	Composed of 5 helical structures, containing 2 subunits, connected by cysteine disulfide bonds.		[52]
	Jug r 1	15–16		Consists of 142 amino acid residues.		[53]
	Jug r 4	58.1	11S globulin	Except for the first 23 amino acid residues which are predicted as signal peptides, the remaining part has a total of 507 amino acid residues.		[54]
	Cor a 9	40		Composed of 515 amino acid residues, the sequence homology with Ara h 3 is about 45%.		[55]
	Pru du 6	350		Exist in the form of hexamers, each monomer subunit is composed of one acid chain of 40 to 42 kDa and one alkaline chain of 20 kDa.		[56]

## Data Availability

Not applicable.

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
