# Peer review of "Aptamer-Based Fluorescent Biosensor for the Rapid and Sensitive Detection of Allergens in Food Matrices"

_foods, 2021, doi:10.3390/foods10112598_

Round 1

Reviewer 1 Report

The manuscript by Hong et al, is a very well designed revision about the detection of food allergens in food. The document is correctly structure and many interesting information is included. The figures illustrates perfectly the methodologies described.

This reviewer recommends focusing the introduction section, specially paragraph 1. Many different and heterogeneous topics are included and all of them are not necessary considering the objective of the manuscript. It would be recommendable to start with the problem and the unmet need of not detecting the allergens. This section should be shortened and focused. Regarding the section where authors described the allergen composition of allergenic extracts, this part should also be shortened and using exclusively the relevant information. The authors should state that other allergens are described. Only a few of them are detected by the methodologies that they describe.

Author Response

Reply: According to comments, we have modified the introduction Section, checked the focus with this review, and made language modification. The rest of the manuscript has been reorganized, removing duplicates (Section 2), which makes it more reasonable and clearer. At present, there are not many studies on aptamer-based fluorescence biosensors for the detection of allergens in food substrates, which means that such detection strategies have not been widely used for the detection of most food allergens. Therefore, in Section 3, we only reviewed studies on adaptive fluorescence sensors that have been successfully applied to actual detection of allergens in food substrates. Relevant modifications have been made, and we hope the enclosed manuscript can meet the requirements and be accepted for publication.

Reviewer 2 Report

General comments: The authors reviewed the development of aptamer-based fluorescent biosensors for food allergen detection. Although the topic is relatively new and potentially of interest to readers, the manuscript is poorly organized and incomprehensive. The entire section 2 on animal and plant food allergens should be removed. The different detection methods should be grouped based on mechanism instead of target allergen. All the figures in this review are from other people’s publications. The authors failed to mention that they have received permission from the publishers to use the figures. Moreover, any irrelevant information in the original figure that is not discussed in the current review should be removed. When discussing the results of each method, the authors only talked about the detection mechanism and sensitivity of the assays. However, other important parameters such as assay specificity, reproducibility, robustness, etc. should also be addressed. The authors should also critically assess the advantages and limitations of the technique. In addition, the manuscript is poorly written. Grammatical errors and awkward syntax are found throughout the manuscript. The authors should consider using a professional language editing service. Additional queries, comments, and suggestions are listed in in my specific comments.

Specific comments

Line 11: “life quality”.

Line 12: Delete “Only strictly”.

Line 14: Change “identify” to “detect”.

Line 16: “targets”.

Line 30: thousands is a gross understatement.

Line 35: If the immune system cannot recognize food allergens, how can it react?

Lines 37-38: Awkward sentence.

Lines 40-41: The authors need to update their reference. The prevalence is much higher now.

Line 42: “fasting” is a wrong word.

Line 56: Confusing sentence.

Lines 95-250: This topic has been reviewed by many and the authors did not provide any new information. Moreover, it is out of the focus of the current review and should be removed.

Line 101: The big eight allergenic foods are only relevant to certain regions.

Line 184: “animal” should be “plant”.

Figure 1 has a poor resolution and should be improved. Did the authors obtain permission from Talanta to use the figure in this review?

Line 309: “arginine kinase”.

Line 322: The authors did not address assay specificity.

Lines 342-344: Summarize the findings.

Line 379 “higher affinity to”.

Line 449: What is “hes”?

Line 482: Awkward sentence.

Author Response

Response to Reviewer 2 Comments

General comments: The authors reviewed the development of aptamer-based fluorescent biosensors for food allergen detection. Although the topic is relatively new and potentially of interest to readers, the manuscript is poorly organized and incomprehensive. The entire section 2 on animal and plant food allergens should be removed. The different detection methods should be grouped based on mechanism instead of target allergen. All the figures in this review are from other people’s publications. The authors failed to mention that they have received permission from the publishers to use the figures. Moreover, any irrelevant information in the original figure that is not discussed in the current review should be removed. When discussing the results of each method, the authors only talked about the detection mechanism and sensitivity of the assays. However, other important parameters such as assay specificity, reproducibility, robustness, etc. should also be addressed. The authors should also critically assess the advantages and limitations of the technique. In addition, the manuscript is poorly written. Grammatical errors and awkward syntax are found throughout the manuscript. The authors should consider using a professional language editing service. Additional queries, comments, and suggestions are listed in in my specific comments.

Reply: Thanks for the review. We have revised the manuscript carefully. Section 2 has summarized the main allergens in animal and plant foods and their related research, which is considered to be a necessary part of the manuscript because most research on aptamer-based fluorescent biosensors for detecting allergens in food matrices have focused on certain allergens. We have further classified the different allergens according to the application of this detection method. According to the comment, the figures that were not fully discussed were deleted and the permission to use all the figures in this article have been obtained and have made relevant descriptions in the figure captions. The specificity of each reported strategy, time required, method comparison, etc. were discussed in detail. In Section 4, we not only summarized the advantages of aptamer-based fluorescent biosensors in detecting allergens in food, but also analyzed some of the challenges encountered in the practical application of this method. We hope the enclosed manuscript can meet the requirements and be accepted for publication of Foods. Thanks for your review.

Specific comments

Line 11: “life quality”. (Line 12: Delete “Only strictly”.) (Line 14: Change “identify” to “detect”.) (Line 16: “targets”.) (Line 30: thousands is a gross understatement.) (Lines 37-38: Awkward sentence.) (Line 42: “fasting” is a wrong word.) (Line 56: Confusing sentence.) (Line 184: “animal” should be “plant”.) (Line 309: “arginine kinase”.) (Line 379 “higher affinity to”.) (Line 449: What is “hes”?) (Line 482: Awkward sentence.)

Reply: According to the comment, we have modified the mentioned part in the revised manuscript (Line 11, 12, 15, 16, 33, 36-38, 36, 48-52, 186, 125, 431, 492, 523-525). We also invited the skilled researchers for language editing of the revised. All the revisions have been marked in color in the manuscript. Thanks for your review.

Line 35: If the immune system cannot recognize food allergens, how can it react?

Reply: Thanks for the comment. When food allergens enter the human body for the first time, the immune system regards them as harmful substances and produces an excessive protective immune response, producing specific antibodies. When exposed to allergens again, these antibodies can specifically recognize and bind to them, stimulate the release of specific bioactive substances, and promote the inflammatory response of various organs or tissues of the body. Thanks for your review.

Lines 40-41: The authors need to update their reference. The prevalence is much higher now.

Reply: According to the comment, we have updated the related data of the revised manuscript (Line 34) according to the latest references. Thanks for your review.

Lines 95-250: This topic has been reviewed by many and the authors did not provide any new information. Moreover, it is out of the focus of the current review and should be removed.

Reply: Thanks for your review. This paper focuses on reviewing the relative research on the types of allergens in food matrices and the application of aptamer-based fluorescent sensors in allergen detection in order to provide effective reference for subsequent allergen related research. The common allergens in different food matrices were introduced in detail in Section 2. Although some researchers have also mentioned this part, we thought this study was more comprehensive and specific. The latest reports in the past five years were referred, which provide the latest information for future studies on allergens. Due to the Section 2 provides a basis for the Section 3, Section 2 is still retained in the revised manuscript. Thanks again for the comment.

Line 101: The big eight allergenic foods are only relevant to certain regions.

Reply: Thanks for the comment. We have referred the relevant references and found that the Food and Agriculture Organization (FAO) of the United Nations listed eight food allergens, namely milk, eggs, fish, crustaceans, peanuts, soybeans, stone fruits category and wheat in 1995. Therefore, we have added this information in the revised manuscript. Please refer to the revised manuscript. Thanks for your review.

Figure 1 has a poor resolution and should be improved. Did the authors obtain permission from Talanta to use the figure in this review?

Reply: We have supplied one clear Figure 1 in the revised manuscript. we have obtained the permission of this figure, which is stated in the Figure 1 caption. Please refer to the revised manuscript. Thanks for the comment.

Line 322: The authors did not address assay specificity.

Reply: Thanks for the comment. Specificity is an important indicator of aptamer-based fluorescence biosensor, which affects its practical application ability. In the revised manuscript, the specificity of various developed biosensors was shown. Please refer to the revised manuscript. Thanks for your review.

Lines 342-344: Summarize the findings.

Reply: According to the comment, we have carefully summarized the finding of this review, and offered a clear revised manuscript. Please refer to the revised manuscript. Thanks for your review.

Reviewer 3 Report

Comments to authors:

This manuscript is well-organized regarding development and application of aptamers for detection of food allergens. English and scientific writings are also good.

Line 50: this sentence is redundant. Need to be revised.

Line 52-54: it is clear that PCR is suitable for detection of DNA fragments, but not of proteins and metabolites, due to the principle of the method. Reference #16 seems to report detection of Salmonella DNA fragments, but not of any food allergens, which is unsuitable for the reference here.

Line 176: At line 170, authors mention that OVM structure contains 20-25% sugar-based component. Does this mean OVM is a glycoprotein? If so, OVT is not the only glycoprotein. Please make sure.

Line 184: This subsection title is the same as that of the subsection 2.1.

Line 228: Double periods.

Line 248: This subsection has no sentence or paragraph. I think the contents of Table 1 are also explained similarly to paragraphs described above.

Line 253: I think it is better for readers to provide more of details of methodology regarding aptamer screening including molecular mechanisms and principle.

Line 319: This sentence is the same as the former sentence and is redundant.

Line 322: reliable rather than dependable ?

Line 348: Please make this sentence more understandable easily for readers.

Line 449: What do you mean by hes?

Line 479: This sentence does not make sense. Please check.

Author Response

Reply to Reviewers

We highly appreciate the constructive comments and suggestions of referees which are very helpful for us to revise the manuscript. Details reply to the comments and suggestions are listed below:

Response to Reviewer 3 Comments

Line 50: this sentence is redundant. Need to be revised. (Line 184: This subsection title is the same as that of the subsection 2.1.) (Line 228: Double periods.) (Line 319: This sentence is the same as the former sentence and is redundant.) (Line 322: reliable rather than dependable?) (Line 348: Please make this sentence more understandable easily for readers.) (Line 449: What do you mean by hes?) (Line 479: This sentence does not make sense. Please check.)

Reply: According to the comment, we have revised the manuscript to ensure that it is more readable. We also invited some skilled researchers to revise the manuscript. All the changes were marked in color in the revised manuscript. We hope the manuscript can meet the requirement. Thanks for your review.

Line 52-54: it is clear that PCR is suitable for detection of DNA fragments, but not of proteins and metabolites, due to the principle of the method. Reference #16 seems to report detection of Salmonella DNA fragments, but not of any food allergens, which is unsuitable for the reference here.

Reply: Thanks for the comment. We have modified this part in the revised manuscript (Line 47, Reference 12). Please refer to the revised manuscript.

Line 176: At line 170, authors mention that OVM structure contains 20-25% sugar-based component. Does this mean OVM is a glycoprotein? If so, OVT is not the only glycoprotein. Please make sure.

Reply: After verification, OVM and OVT are glycoproteins. We have modified the relevant content in the revised manuscript. Please refer to the revised manuscript (lines 168-185).

Line 248: This subsection has no sentence or paragraph. I think the contents of Table 1 are also explained similarly to paragraphs described above.

Reply: Thanks for your review. Table 1 has illustrated the major allergens in food matrices and their allergenic properties. Section 2 introduces the detailed information of various allergens. The contents of the two are not interchangeable. Now, Table 1 has been adjusted to an appropriate position of the revised manuscript. According to the comment, the repetitive content has been modified to make this part more reasonable. Thanks for your review.

Line 253: I think it is better for readers to provide more of details of methodology regarding aptamer screening including molecular mechanisms and principle.

Reply: In this paper, the principle and procedure of aptamer screening was introduced, and the binding mechanism of aptamer and target was emphasized, which played a key role in the application of aptamer sensors in actual samples. According to the comment, the aptamer-target interaction mechanism was supplied in Section 3.2. Please refer to the revised manuscript. Thanks for your review.

Round 2

Reviewer 2 Report

General comments: The authors addressed most of my comments and revised their manuscript accordingly. I still have a few comments and suggestions.

Specific comments

Line 13: Delete “and treat”. Allergen avoidance is not a treatment.

Line 44: Since reference #11 reported an ELISA for inhalant allergen, it should be replaced by a reference that reports ELISA for food allergen detection, such as the one below:

Liu, C., Zaffran, V. D., Gupta, S., Roux, K. H., & Sathe, S. K. (2019). Pecan (Carya illinoinensis) detection using a monoclonal antibody-based direct sandwich enzyme-linked immunosorbent assay. LWT, 116, 108516.

Line 88: “A great variety of food allergens exist widely in nature”.

Table 1: Include IUIS allergen names for all the listed allergens when available.

Table 1: This table does not seem to be comprehensive. For example, the 11S globulin allergens are missing in the nuts category.

Line 136: Delete “healthy”.

Line 159: Change “will” to “may” since not all people can outgrow egg allergy..

Line 192: Delete “allergens”.

Author Response

Response to Reviewer 2 Comments

Specific comments

Line 13: Delete “and treat”. Allergen avoidance is not a treatment. (Line 88: “A great variety of food allergens exist widely in nature”.) (Line 136: Delete “healthy”.) (Line 159: Change “will” to “may” since not all people can outgrow egg allergy.) (Line 192: Delete “allergens”)

Reply: According to the comment, we have revised the manuscript to ensure that it is more readable. All the changes were marked in red in the revised manuscript. We hope the manuscript can meet the requirement. Thanks for your review.

Line 44: Since reference #11 reported an ELISA for inhalant allergen, it should be replaced by a reference that reports ELISA for food allergen detection, such as the one below:

Liu, C., Zaffran, V. D., Gupta, S., Roux, K. H., & Sathe, S. K. (2019). Pecan (Carya illinoinensis) detection using a monoclonal antibody-based direct sandwich enzyme-linked immunosorbent assay. LWT, 116, 108516.

Reply: Thanks for your review, we have modified this part in the revised manuscript (Line 45, Reference 11). Please refer to the revised manuscript.

Table 1: Include IUIS allergen names for all the listed allergens when available.

Reply: According to the comment, we have added the IUIS allergen names for all the listed allergens in Table 1. Please refer to the revised manuscript. Thanks for your review.

Table 1: This table does not seem to be comprehensive. For example, the 11S globulin allergens are missing in the nuts category.

Reply: Thanks for your review. We have supplemented the 11S globulin allergens in nuts in the corresponding positions in Table 1. Please refer to the revised manuscript. Thanks again for the comment.